# Genome wide analysis for mouth ulcers identifies associations at immune regulatory loci

Tom Dudding [1,2], Simon Haworth [1,2], Penelope A. Lind [3], J. Fah Sathirapongsasuti[4], the 23andMe Research Team[#], Joyce Y. Tung[4], Ruth Mitchell [1], Lucía Colodro-Conde [3], Sarah E. Medland[3], Scott Gordon [5], Benjamin Elsworth [1], Lavinia Paternoster [1], Paul W. Franks[6,7,8], Steven J. Thomas[2], Nicholas G. Martin [5] & Nicholas J. Timpson[1]

Mouth ulcers are the most common ulcerative condition and encompass several clinical diagnoses, including recurrent aphthous stomatitis (RAS). Despite previous evidence for heritability, it is not clear which specific genetic loci are implicated in RAS. In this genome-wide association study ($n = 461,106$) heritability is estimated at 8.2% (95% CI: 6.4%, 9.9%). This study finds 97 variants which alter the odds of developing non-specific mouth ulcers and replicate these in an independent cohort ($n = 355,744$) (lead variant after meta-analysis: rs76830965, near *IL12A*, OR 0.72 (95% CI: 0.71, 0.73); $P = 4.4\text{e}{-}483$). Additional effect estimates from three independent cohorts with more specific phenotyping and specific study characteristics support many of these findings. In silico functional analyses provide evidence for a role of T cell regulation in the aetiology of mouth ulcers. These results provide novel insight into the pathogenesis of a common, important condition.

[1] Medical Research Council Integrative Epidemiology Unit, Department of Population Health Sciences, Bristol Medical School, University of Bristol, Bristol BS8 2BN, UK. [2] Bristol Dental School, University of Bristol, Bristol BS1 2LY, UK. [3] Department of Psychiatric Genetics, QIMR Berghofer Medical Research Institute, Brisbane 4006 Queensland, Australia. [4] Research, 23andMe, Inc, Mountain View 94041 CA, USA. [5] Department of Genetic Epidemiology, QIMR Berghofer Medical Research Institute, Brisbane 4006 Queensland, Australia. [6] Genetic and Molecular Epidemiology Unit, Department of Clinical Sciences, Lund University, Malmö 221 00, Sweden. [7] Department of Public Health & Clinical Medicine, Umeå University, Umeå 901 87, Sweden. [8] Department of Nutrition, Harvard T.H. Chan School of Public Health, Harvard University, Boston 02115 MA, USA. These authors contributed equally: Tom Dudding, Simon Haworth. [#]A full list of consortium members appears at the end of the paper. Correspondence and requests for materials should be addressed to N.J.T. (email: N.J.Timpson@bristol.ac.uk)

Oral ulceration is the most common ulcerative condition in humans, affecting up to 25% of young adults[1] and a higher proportion of children[2]. Depending on context, ulcers in the mouth are described as mouth ulcers or canker sores, both of which are descriptive terms rather than clinical diagnoses. In this article, mouth ulcers is used as an umbrella term describing the spectrum of clinical presentation.

To date, many causes of mouth ulcers are recognised including mucosal trauma and a range of autoimmune and inflammatory conditions. For example, mouth ulcers are common in patients with ulcerative colitis and Crohn's disease[3], systemic lupus erythematosus[4,5] and are considered a diagnostic feature of Behçet's disease, an inflammatory disorder of blood vessels that causes ulceration of the mouth, eyes and genitals[6].

Many people experience recurrent mouth ulcers which cannot be attributed to systemic disease or obvious oral trauma. Here, the clinical diagnosis of recurrent aphthous stomatitis (RAS) is used, referring to a group of closely related conditions of uncertain aetiology, whose defining feature is the presence of clinically characteristic oral ulcers, which are painful and associated with impaired quality of life[7–9]. First line management strategies for RAS reduce the severity of ulceration without preventing recurrence[10]. Second line management strategies involve topical or systemic use of potent and non-specific immunomodulatory drugs including thalidomide and dapsone, exposing patients to a range of side effects[11,12]. Thus, there is unmet need for a wider range of therapeutic options in the management of RAS.

In part, the lack of satisfactory management for RAS reflects uncertainty in the exact aetiology. A number of predisposing factors have been reported including vitamin or haematinic deficiency, chemicals such as sodium laurel sulphate, mechanical trauma, stress and anxiety or infection with bacteria or viruses[1,13]. Regardless of the initial trigger, it is believed that immune regulation plays a pivotal role in mediating tissue damage and the clinical presentation of RAS[1]. Susceptible individuals experience focal infiltration of the oral mucosa by monocytes and T lymphocytes deep to the basal membrane, followed by loss of superficial mucosa and a proliferative healing phase[14,15].

Family-based studies support a role of genetic susceptibly in the aetiology of RAS[2,16,17] yet to date the genetic basis of this susceptibility remains poorly characterised. Previous candidate gene association studies have investigated variation in the region of genes encoding key cytokines (TNF-α, IL-1α, IL-1β, IL-6, IL-10, IL-12), with varied results[18–21]. Genome-wide association studies (GWAS) have the potential to identify genetic variants associated with both susceptibility to initial triggers and the immune reaction that leads to the tissue damage and ulcer formation.

A recent study used genome-wide data to look for associations between gene pathways and specific sub-phenotypic features (including mouth ulcers) in a case series of patients with systemic lupus erythematosus[22]. The study found some evidence for an association between the vascular endothelial growth factor (VEGF) pathway and the oral ulcer sub-phenotype. It is not clear how these findings relate to the general population and there are no previous conventional genome-wide association studies for RAS or mouth ulcers.

There is therefore a need to undertake genome-wide analysis for RAS, but specific measures are unavailable in large cohorts. The most appropriate strategy given available data, is to use large collections to identify and replicate associated genetic variants in a well-powered GWAS of self-reported non-specific mouth ulcers and then validate the effects of these variants in smaller collections with more clinically relevant RAS-specific measures. Given that RAS is nested within mouth ulcers, the inclusion of other causes of ulcers would introduce, at worst, noise into the analysis and, at best, enhance it by highlighting mechanisms which are relevant to oral mucosal breakdown irrespective of trigger.

This genome-wide association study identifies 97 variants which alter the odds of developing non-specific mouth ulcers and replicates them in an independent cohort. In silico functional analyses provide evidence for a role of T cell regulation in the aetiology of mouth ulcers. These results provide novel insight into the pathogenesis of a common, important condition.

## Results

**Contributing studies.** In UK Biobank (UKBB) participants with data on mouth ulcers, the mean age at questionnaire completion was 56.7 years (range = 38.0, 73.0), 54.2% of participants were female and 10.2% of participants reported having mouth ulcers within the last year. In research participants from the personal genetics company 23andMe, Inc. with data on canker sores, 67.8% were over 45 years old, 59.2% of participants were female and 72.4% of participants reported ever having canker sores. In the QIMR Berghofer Medical Research Institute (QIMR) Over 50s (Aged) study (AG), which examined aging and age-related disease in twin pairs from the Australian Twin Registry, the mean age was 61.2 years (range = 50.2, 85.6). A large proportion of participants were female (74.1%) and the proportion reporting a history of mouth ulcers was 18.7%. Two other studies exclusively or partially included younger participants, the Avon Longitudinal Study of Parents and Children (ALSPAC), a UK population-based birth cohort (mean age = 23.9 years, range = 22.8, 25.3; 65.2% females), and QIMR Melanocytic Naevi in Adolescent Twins (TW) study, which primarily examined melanotic naevae in twin pairs (mean age 24.2 years, range = 10.1, 62.3 (a combination of adolescents and their parents); 53.8% females). In these studies, the proportion reporting ulcers was much higher (ALSPAC: ulcer cases = 74%, TW: percentage reporting having ulcers at least rarely = 86.8%) (Table 1).

**Genome-wide discovery analysis for mouth ulcers.** The primary genome-wide analysis was undertaken in UK Biobank. At an aggregate, genome-wide level there was evidence for a genetic contribution to mouth ulcers, with heritability estimated at 8.2% (95% CI: 6.4%, 9.9%) under an infinitesimal model implemented in linkage disequilibrium score regression (LDSR)[23]. Under a non-infinitesimal model implemented in Heritability Estimator from Summary Statistics (HESS)[24], heritability was estimated at 8.7% (95% CI: 8.2%, 9.2%). Genomic inflation factor (lambda GC) was estimated at 1.20. The intercept term from univariate LDSR was 1.03 (95% CI: 1.01, 1.05) suggesting that most inflation in lambda GC was attributable to polygenicity rather than bias.

After final quality control (QC), 9,851,866 genetic variants were included in GWAS. Evidence for genome-wide association ($P < 5e−8$) with mouth ulcers was seen at 7127 single variants (Fig. 1, Supplementary Figure 1). After clumping, using a LD threshold ($r^2 = 0.1$) in PLINK, these formed 97 approximately independent lead variants.

**Signal validation.** All 97 of these independent variants showed directional consistency in 23andMe, with comparable effect sizes in both collections (Supplementary Figure 2). Summary statistics for the 10 strongest associated variants after meta-analysis of UK Biobank and 23andMe are shown in Table 2. All 97 variants are shown in Supplementary Data 1.

**Assessment of effect sizes in additional populations.** Three smaller samples were used to assess the effect sizes of the 97 lead

## Table 1 Demographics of samples included in analysis

| Study (short name) | N (genotype and phenotype data) | N (%) Cases | Severity level n (%) | | | | Mean age [range]ᵃ | Proportion female (%) |
|---|---|---|---|---|---|---|---|---|
| | | | Never | Rarely | Sometimes | Frequently | | |
| UKBB | 461,106 | 47,079 (10.2) | | | | | 56.7 [38.0, 73.0] | 54.2 |
| 23andMe | 355,744 | 98,298 (72.4) | | | | | 51.1 [36.0, 66.0]ᵃ | 59.2 |
| ALSPAC | 2976 | 2201 (74.0) | | | | | 23.9 [22.8, 25.3] | 65.2 |
| AG | 1120 | 209 (18.7) | | | | | 61.2 [50.2, 85.6] | 74.1 |
| TW | 2442 | | 115 (13.2) | 406 (46.7) | 283 (32.5) | 66 (7.6) | 24.2 [10.1, 62.3] | 53.8 |

*UKBB* UK Biobank, *ALSPAC* Avon Longitudinal Study of Parents and Children, *AG* QIMR Berghofer Medical Research Institute's (Aged) study', *TW* QIMR Berghofer Medical Research Institute Melanocytic Naevi in Adolescent Twins study
ᵃFor 23andMe the interquartile range is given instead of the range

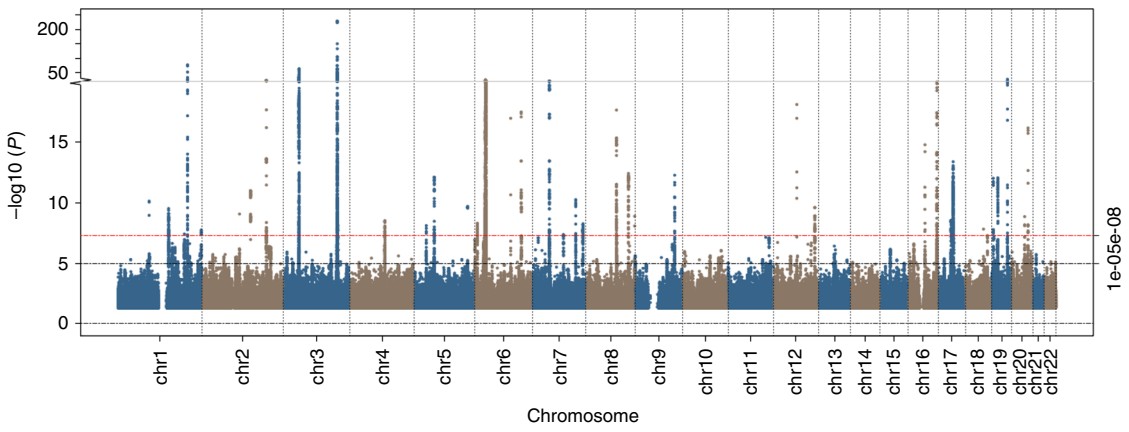

**Fig. 1** Manhattan plot of genome-wide association analysis of self-reported ulcers in UK Biobank

## Table 2 Ten lead single variant results from GWAS in UK Biobank, replication in 23andMe and after Meta-analysis

| Variant | Chr | Position | Effect allele | Other allele | UK Biobank (N = 461,106) | | | 23andMe (N = 355,744) | | Meta-analysis (N = 816,850) | |
|---|---|---|---|---|---|---|---|---|---|---|---|
| | | | | | EAF | Odds ratio (95% CI) | P-value | Odds ratio (95% CI) | P-value | Odds ratio (95% CI) | P-value |
| rs1800871 | 1 | 206946634 | A | G | 0.22 | 1.17 (1.15, 1.18) | 2.10e−77 | 1.19 (1.18, 1.21) | 3.51e−164 | 1.18 (1.17, 1.19) | 6.05e−236 |
| rs4683205 | 3 | 46334670 | G | A | 0.47 | 1.12 (1.11, 1.14) | 5.70e−64 | 1.08 (1.07, 1.09) | 4.07e−48 | 1.10 (1.09, 1.11) | 4.94e−106 |
| rs34390431 | 3 | 46464940 | G | A | 0.35 | 0.92 (0.90, 0.93) | 1.90e−32 | 0.94 (0.93, 0.95) | 3.11e−25 | 0.93 (0.92, 0.94) | 2.90e−54 |
| rs11928736 | 3 | 159565409 | G | C | 0.56 | 1.08 (1.06, 1.09) | 3.40e−36 | 1.07 (1.06, 1.08) | 6.20e−36 | 1.07 (1.06, 1.08) | 2.62e−60 |
| rs76830965 | 3 | 159637678 | C | A | 0.88 | 0.71 (0.69, 0.72) | 1.60e−229 | 0.73 (0.72, 0.74) | 3.06e−268 | 0.72 (0.71, 0.73) | 4.4e−483 |
| rs7645203 | 3 | 159686669 | C | T | 0.60 | 1.09 (1.07, 1.10) | 4.20e−33 | 1.08 (1.07, 1.09) | 1.96e−48 | 1.08 (1.08, 1.09) | 9.65e−80 |
| rs55667203 | 3 | 159950798 | C | T | 0.83 | 0.91 (0.89, 0.92) | 2.60e−25 | 0.90 (0.89, 0.92) | 1.96e−41 | 0.91 (0.90, 0.92) | 1.51e−64 |
| rs2523589 | 6 | 31327334 | G | T | 0.50 | 0.93 (0.92, 0.94) | 2.70e−26 | 0.94 (0.93, 0.95) | 2.56e−30 | 0.94 (0.93, 0.94) | 1.63e−54 |
| rs7749390 | 6 | 137540370 | A | G | 0.62 | 1.06 (1.05, 1.08) | 3.30e−18 | 1.08 (1.07, 1.09) | 1.03e−46 | 1.08 (1.07, 1.08) | 1.98e−62 |
| rs3764613 | 19 | 46896217 | A | G | 0.43 | 0.93 (0.91, 0.94) | 1.10e−28 | 0.93 (0.92, 0.94) | 6.82e−46 | 0.93 (0.92, 0.93) | 7.41e−73 |

All 97 variants reaching genome-wide significance given in Supplementary Data 1. Base pair positions are given with reference to build 37 of human genome reference consortium
*Chr* chromosome, *CI* confidence interval, *EAF* effect allele frequency in UK Biobank

variants with more RAS-specific phenotypes. In the AG and TW studies, clinical photographs were used to help participants identify RAS as opposed to traumatic ulcers. In the TW study, measures of ulcer severity were collected and the adolescents within the study (1572 of 2442 participants) were near the age of highest RAS prevalence. In the ALSPAC study all participants were in this high RAS prevalence age range. Twenty-four of the 97 lead variants showed consistent effect direction across all phenotypes from the three independent collections (Supplementary Data 1).

**Description of lead novel associations.** The strongest evidence for association, after meta-analysis of UK Biobank and 23andMe, was seen at rs76830965, a common variant lying within *ILAS1-AS1* ~69 kb 5′ of *IL12A* on chromosome 3. This variant conferred large effects on the odds of mouth ulcers (odds ratio (OR)

0.72 per C allele, 95% CI: 0.71, 0.73; effect allele frequency (EAF) 0.89; $\chi^2$ test $P = 4.4e{-}483$). Complementary evidence for a protective effect of the C allele was seen in all three lookup cohorts (ALSPAC: OR 0.67, 95% CI: 0.56, 0.79; $P = 4.0e{-}06$, AG: OR 0.75, 95% CI: 0.53, 1.07; $P = 0.11$ and TW: OR 0.86, 95% CI: 0.77, 0.96; $P = 7.8e{-}03$) (Table 2, Fig. 2a). After clumping, other variants in the same 3q25 locus as rs76830965 showed very strong evidence for association and showed consistent effects in all but the AG cohort. For example, rs7645203 2.0 kb 5′ of *IL12A* (OR 1.08 per C allele, 95% CI: 1.08, 1.09; EAF 0.59; $P = 9.6e{-}80$), rs11928736 within *SCHIP1* (OR 1.07 per G allele, 95% CI: 1.06, 1.08; EAF 0.56; $P = 2.62e{-}60$), and rs55667203 within *RP11-432B6.3* (OR 0.91 per C allele, 95% CI: 0.90, 0.92; EAF 0.84; $P = 1.5e{-}64$) (Table 2).

rs1800871, a variant within 1 kb 5′ of *IL10*, showed the second strongest evidence for association after meta-analysis, conferring

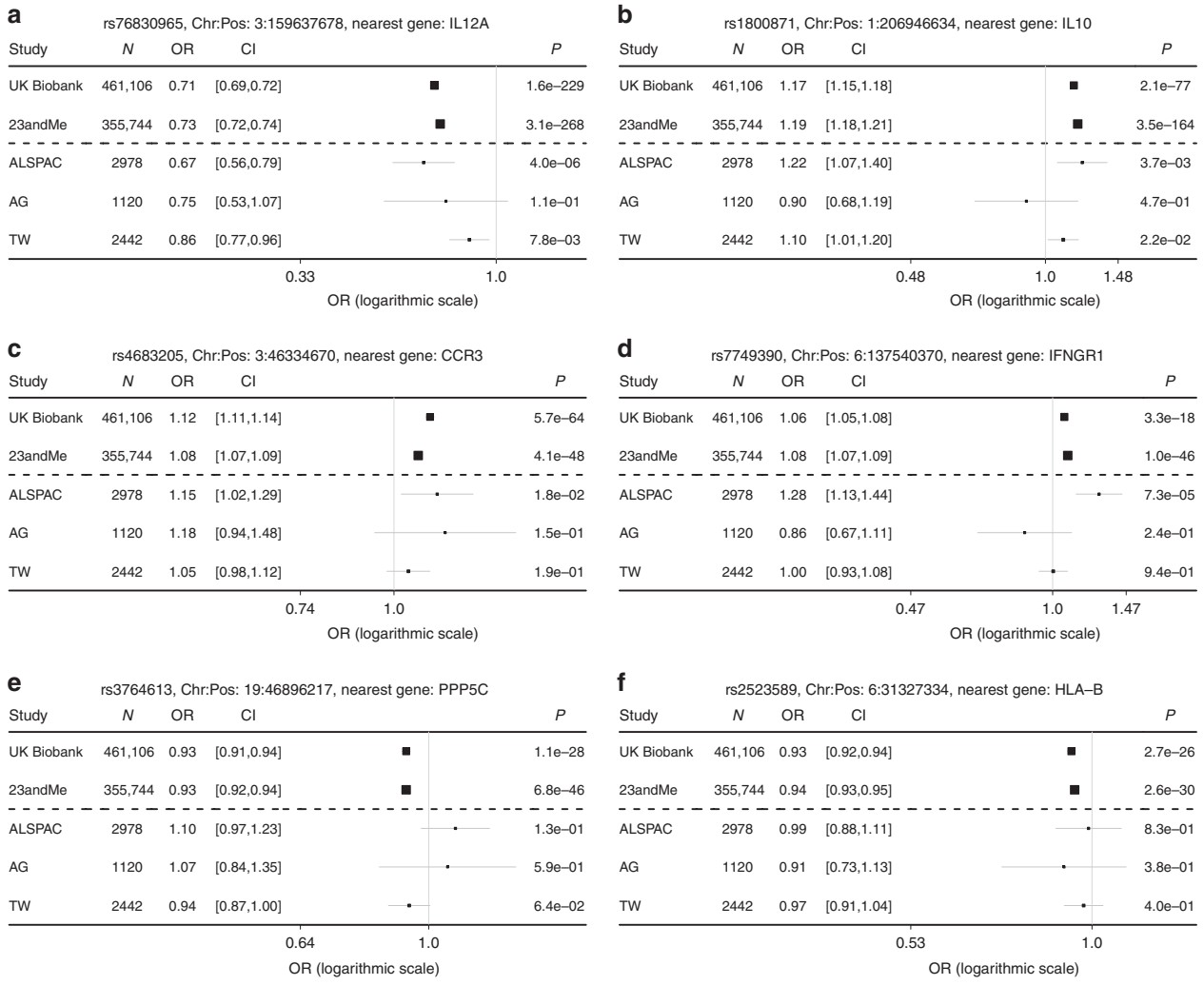

**Fig. 2** Forest plot detailing estimates from GWAS, replication and lookup samples for selected variants. Each panel (**a–f**) represents the effect size per allele of the variant on the mouth ulcer phenotype for that study. Estimates are odds ratios and errors bars indicate the 95% confidence interval. UK Biobank, ALSPAC and AG = ulcer case control phenotype, TW = severity ulcer phenotype. Chr = Chromosome, Pos = Genomic position (build 37)

a large effect on mouth ulcers (OR 1.18 per A allele, 95% CI: 1.17, 1.19; EAF 0.22; $P = 6.0e-236$) (Table 2, Fig. 2b).

In the 3p21 locus, rs4683205 near *CCR3* showed strong evidence for a moderate effect on mouth ulcers (OR 1.10, 95% CI: 1.09, 1.11; EAF 0.47; $P = 4.9e-106$) (Table 2, Fig. 2c). Additional variants in this region (e.g. rs4493469 near *CCR3* and rs34390431 near *CCRL2*) also showed strong evidence for an association with mouth ulcers after clumping (Supplementary Data 1).

Other associated common variants showed more modest effects on mouth ulcers. For example, rs7749390 an intronic variant in the *IFNGR1* gene (Table 2, Fig. 2d) and rs3764613 within *PPP5C* (Table 2, Fig. 2e), showed very strong evidence for associations with mouth ulcers (OR 1.08, 95% CI: 1.07, 1.08; EAF 0.61; $P = 2.0e-62$ and OR 0.93, 95% CI: 0.92, 0.93; EAF 0.43; $P = 7.4e-73$, respectively).

Within the HLA (chromosome 6p21) there was very strong evidence for association between rs2523589 (2.4 kb 5′ of *HLA-B*) and the odds of developing mouth ulcers (OR 0.94 per G allele, 95% CI: 0.93, 0.94; EAF 0.50; $P = 1.6e-54$). The estimated effect size was moderate and showed directional consistency in all three lookup cohorts (Fig. 2f). Additional haplotype analyses were undertaken to characterise this association, as described under sensitivity analyses, below.

**Gene prioritisation and enrichment analyses.** Gene prioritisation analysis in DEPICT was performed to nominate plausible biologically causal genes by identifying genes in different statistically associated loci ($P < 5e-8$ after clumping) that have similar predicted functions more often than expected by chance. This analysis suggested genes encoding chemokines or chemokine receptors were plausible candidates at many associated loci, for example suggesting *IL12A* at the lead associated chr3:159483176-160796695 locus and *IL12B* at the chr5:158741791–158757895 locus. In loci containing multiple genes with related functions, several potential candidates were identified; for example, association at chr3:45864808–46621589 produced a number of candidates including *CCR1, CCR2, CCRL2, CCR3,* and *CCR5* among others (Supplementary Data 2).

DEPICT assesses whether any of 14,461 pre-computed gene sets are enriched for genes in the associated loci more than would be expected by a randomly distributed phenotype. This analysis identified enrichment in 895 sets with a false discovery rate < 0.01. The strongest statistical evidence for enrichment was seen for MP:0008560 (increased tumour necrosis factor secretion; Z-test $P = 9.74e-12$). Strong evidence for enrichment in several T cell regulatory gene sets was observed, for example, GO:0046632 (alpha-beta T cell differentiation; $P = 2.00e-08$), GO:0046634

(regulation of alpha-beta T cell activation; $P = 2.70e-08$), GO:200514 (regulation of CD4$^+$, alpha-beta T cells; $P = 2.88e-08$), GO:0002286 (T cell activation involved in immune response; $P = 3.75e-08$), GO;0042098 (T cell proliferation; $P = 3.87e-08$) and GO:0045580 (regulation of T cell differentiation; ($P = 4.43e-08$) amongst others (Supplementary Data 3).

Furthermore, DEPICT assesses whether genes in associated loci are highly expressed in any of 209 tissue/cell type annotations. This revealed enrichment in 36 tissues with false discovery rate < 0.01, with the strongest evidence for enrichment in haemic and immune cell lines. The most robust single finding was evidence for enrichment in gene expression in leucocytes ($P = 2.69e-10$) (Supplementary Data 4).

**Enrichment in regulatory motifs**. Non-parametric enrichment analysis in GARFIELD[25] that accounts for LD, minor allele frequency, matched genotyping variants and local gene density, identified enrichment in genic annotations and tissue-specific annotations that are present more than would be expected by chance. Single variants associated with mouth ulcers were enriched for five prime untranslated region variants by ~18-fold compared to permuted matched controls, suggesting the variants identified in this study may predominantly regulate transcription rather than altering protein structure. Associated variants had 14–18-fold enrichment in DNAse1 hypersensitive sites in a number of T cell lineages, including CD8$^+$ primary cells, CD4$^+$ primary cells, T helper (Th) 1 cells and Th2 cells. These results suggest active gene expression occurs near associated variants in a tissue-specific manner (Supplementary Figures 3–10).

**Imputed gene transcription levels**. Tests for association between ulcers and predicted gene expression were performed using S-PrediXcan[26], which uses predictive models to impute transcript expression levels trained in 48 gene–tissue expression project (GTeX)[27] tissues and then uses full GWAS summary statistics to test for associations between these predicted expression levels and phenotype. Results were then assimilated using the S-MulTiXcan[28] method, which integrates information from multiple tissue-specific predictions to improve statistical power. In total, 25,839 gene transcripts were tested for association, of which 244 transcripts passed a Bonferroni-corrected multiple testing threshold ($P < 1.9e-06$). The strongest evidence for association with mouth ulcers across all tissues was at *IL12A* mirroring the single variant results, with an increase in expression predicted to increase the odds of mouth ulcers (Z-test $P = 2.23e-103$). Other single variant results were mirrored in the results of this analysis with increased expression of *SCHIP1* increasing the odds of mouth ulcers and *IL10* decreasing the odds of mouth ulcers ($P = 8.99e-70$ and $5.60e-55$, respectively). The results for all genes are shown in Fig. 3.

**Tests for genetic correlation**. To assess whether the genetic variants contributing to the heritability of mouth ulcers also influenced other traits, genetic correlations were calculated against publicly available GWAS summary statistics[29]. Genetic correlation results were available for 222 traits, of which two passed a Bonferroni corrected $P$-value threshold of $P < 2.3e-04$. These were neuroticism (rg = 0.23, $P = 1.80e-08$) and depressive symptoms (rg = 0.24, Z-test $P = 5.73e-07$) (Fig. 4). A full list of results is included in Supplementary Data 5. For neuroticism and depressive symptoms, the genetic correlation was further examined using the rho-HESS approach which estimates local genetic correlation between mouth ulcers and these traits. At an aggregate level (i.e. incorporating all common genetic variation in the genome), the rg estimates from rho-HESS gave consistent

interpretation with those from LDSR (neuroticism: rg = 0.18, $P = 8.43e-107$; depressive symptoms: rg = 0.33, $P = 4.16e-21$). Additionally, it shows that genetic correlation between mouth ulcers and these traits is evenly distributed across the genome, without peaks in genetic correlation corresponding to peaks in local heritability of either mouth ulcers or these two traits (Supplementary Data 6 and 7).

**HLA haplotype analysis**. To characterise the association signal seen in single variant results near *HLA-B*, analysis of imputed haplotypes was performed within UK Biobank ($n = 336,038$). This identified 24 haplotypes which were associated with mouth ulcers at a Bonferroni-corrected $P$-value threshold of 0.05. The most robust finding was DRB1*0103, an uncommon haplotype (frequency in controls = 0.017, frequency in cases = 0.022) which was associated with markedly increased odds of mouth ulcers in a fully adjusted logistic regression model (OR = 1.33, 95% CI: 1.26, 1.41; $\chi^2$ $P = 2.03e-24$) (Supplementary Table 1, Supplementary Figure 11).

**Polygenic risk score (PRS) analysis**. A PRS approach was used to examine the ability of a series of PRS for mouth ulcers to predict phenotypic variance in the two QIMR samples (TW and AG, combined adults and adolescents). Using Genome-wide Complex Trait Analysis[30] to control for genetic relatedness in linear mixed models (LMMs) in the predictions, up to 0.37% of variation in mouth ulcer severity ($n = 2442$) and 0.86% in mouth ulcer case status ($n = 3562$) could be accounted for using scores trained in UK Biobank. While scores trained only from loci meeting genome-wide significance in UKB accounted for 0.73% of variation in mouth ulcer severity and 0.21% in mouth ulcer case status. (Supplementary Tables 2 and 3).

**Drug repurposing**. To assess whether associated loci might represent targets for repurposed drug interventions we examined the Open Targets database for pharmacological interventions which might recapitulate the effects of naturally occurring genetic variation. Of the 244 gene transcripts that passed Bonferroni correction in S-PrediXcan, 27 were not recognised by the platform. As the platform limits the number of genes to 200, the 17 with the weakest evidence for association in S-PrediXcan were not included in the model. Fifty-two drugs were identified as potential targets (Supplementary Table 4). Fourteen of these are in phase IV trials including Ustekinumab, an antibody against the IL12 protein, encoded by the IL12A gene.

## Discussion

This large-scale genome-wide association study used a non-specific measure of mouth ulcers finding that, in common with studies of specific ulcer types, mouth ulcers are partly heritable[2,16,17]. Although the estimate of heritability from this study is likely an under-estimate and only provides the lower bound, it is substantially less than the heritability previously estimated in twin studies[16]. Some of this heritability is attributable to genetic variants with large effects on the odds of developing mouth ulcers, such as the lead variant identified here (rs76830965). Effect estimates in samples with RAS-specific phenotypes provide further evidence that these variants are associated with this specific type of oral ulceration. These variants with large effects are closely related to biological mechanisms thought to be relevant to the formation of mouth ulcers. The remainder of the heritability is likely driven by indirect effects of large numbers of variants with modest effects on the odds of developing mouth ulcers. Given these genetic variants, which act indirectly, likely also contribute to a wide range of distal

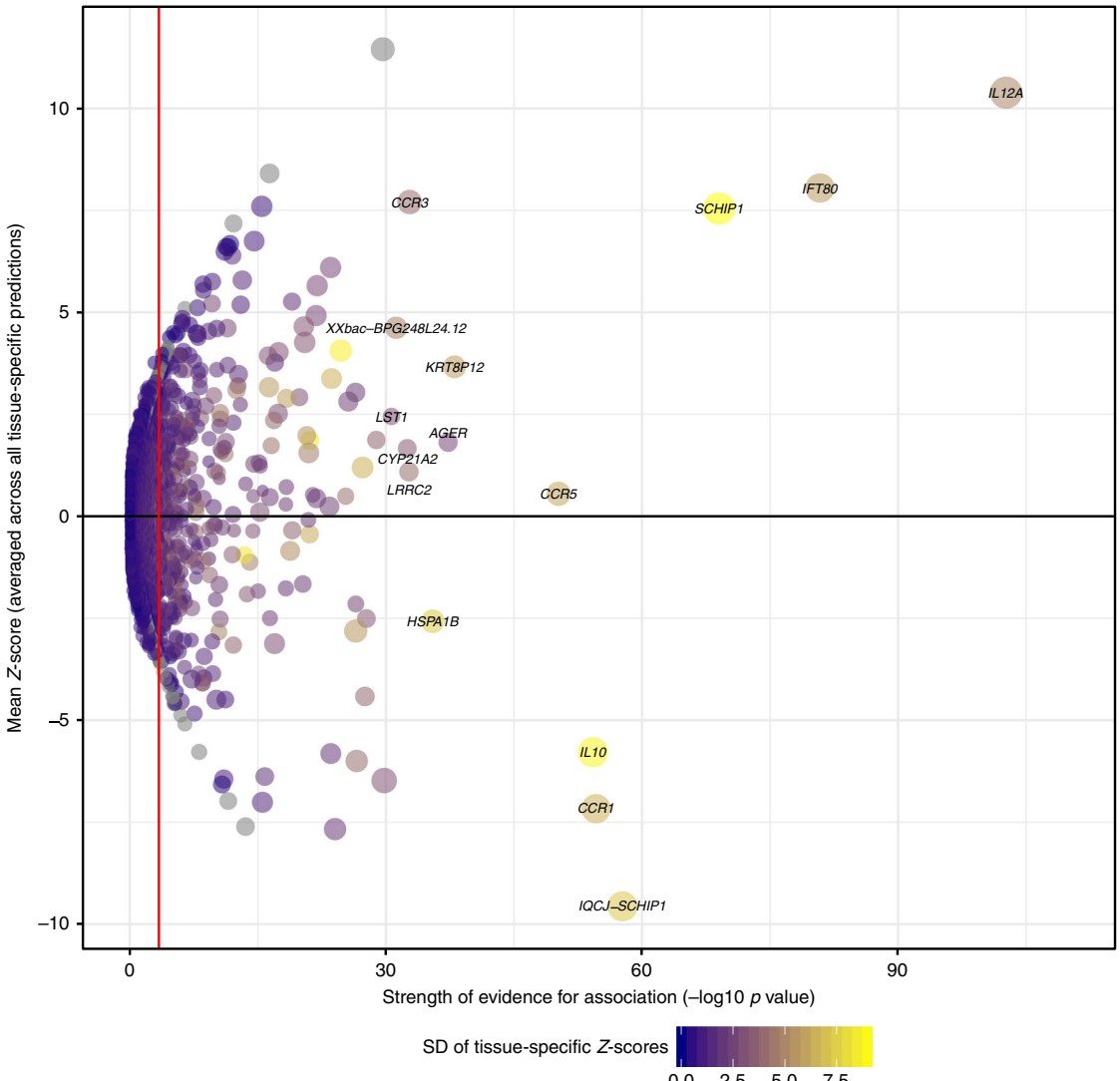

**Fig. 3** Effect of predicted increased transcription of all genes on mouth ulcers. Each dot represented the effect of increased transcription (averaged across all tissue-specific predictions using MultiXcan) on mouth ulcers. The size of the dot indicates the largest effect size in any tissue. The standard deviation (SD) of tissue-specific Z-score is an indicator of tissue specificity with high values (yellow) indicating higher tissue specificity

health-related phenotypes, it is perhaps unsurprising that neuroticism and depressive symptoms show detectable genetic correlation with mouth ulcers in both the LDSR and rho-HESS analyses. The view that this correlation is driven by non-specific associations across the genome, rather than effects at lead-associated variants, is supported by the location specific genetic correlation performed in rho-HESS.

The variants with larger effects on the odds of mouth ulcers are likely to be clinically informative. Many are located in or near genes or are enriched in pathways relating to T cell immunity, and tend to impose a Th1-type immune response, a biologically plausible mechanism that supports previous literature and resonates with clinical practice. Complementary evidence of these variants having true causal associations with RAS is provided by the consistent effect directions seen for many of them in the three independent look-up cohorts with more specific phenotypes.

PRSs, generated using variants selected with a range of significance thresholds, explained only a small amount of variance in the ulcer phenotypes. At present, these PRSs are unlikely to be a clinically useful tool in predicting which patients are at risk of developing mouth ulcers or in predicting severity of mouth ulcers.

Association in the HLA region at the single-SNP level was recapitulated by haplotype analysis, which identified multiple HLA haplotypes which are associated with odds of mouth ulcers. The most striking finding was an association between the DRB1*0103 haplotype and increased odds of mouth ulcers. DRB1*0103 is an uncommon haplotype of *HLA-DRB1* which encodes the beta chain of the HLA-DR heterodimer, forming a ligand for the T cell receptor.

There are traits that commonly present with similar clinical symptoms to mouth ulcers, and the loci identified in this study show commonality with previous GWAS of these traits. Behçet's disease is thought to relate to inappropriate T-cell-mediated inflammatory response and presents clinically with mouth ulcers among other features[31]. Genetic variants at *IL12A, IL10, STAT4, RIPK2, IRF8,* and *CEBPB-PTPN1* which have been reported in previous GWAS for Behçet's disease[32–34], are in high LD and have consistent effect direction with those reported here. This raises the possibility of a similar mechanism leading to clinical presentation with mouth ulcers in both conditions. Elsewhere in the literature there is striking overlap with the genetics of coeliac disease, where T cell-mediated responses are also believed to be important[35]. In particular, rs17810546 (a variant in close LD with

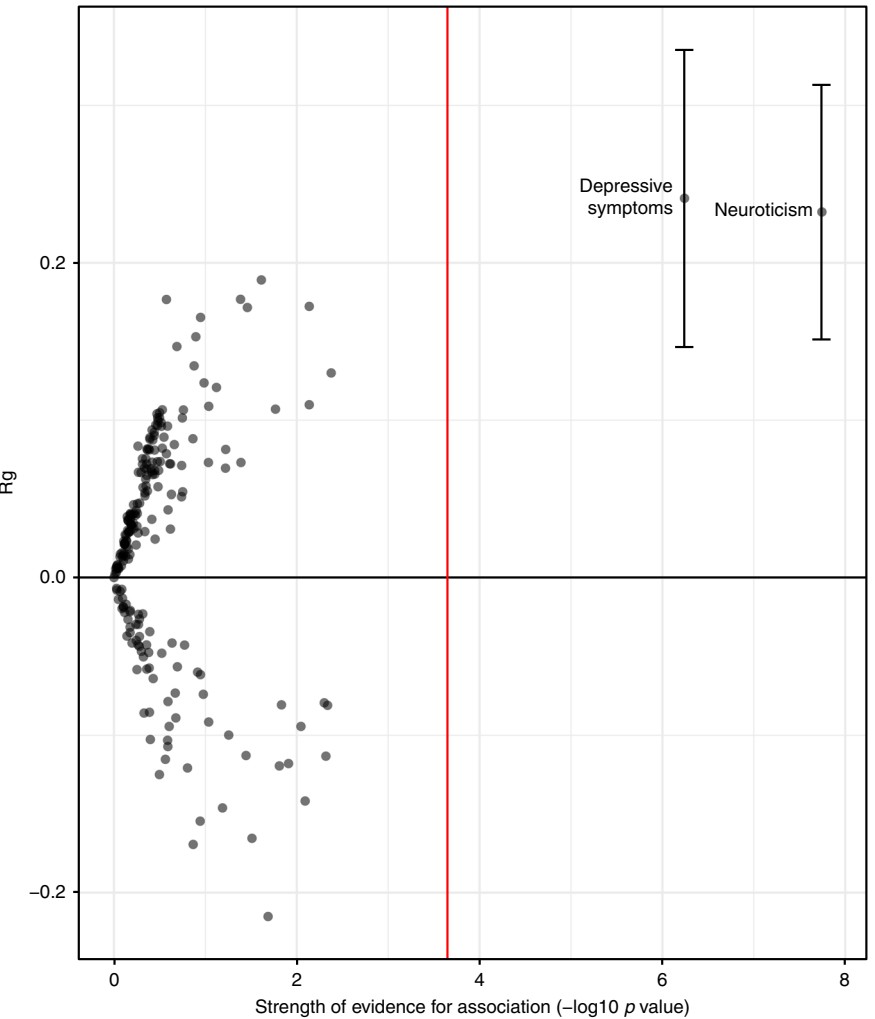

**Fig. 4** Genetic correlation between mouth ulcers and 222 traits. Each dot represented the Rg between mouth ulcers and an individual trait. The red line represents the Bonferroni-corrected multiple testing threshold at the 5% level. Error bars showing 95% confidence intervals and labels are included for traits that pass the multiple testing threshold

rs76830965, the top associated variant in our analysis, $r^2 = 0.98$) was reported as early as 2008 to have large effects on coeliac disease[36], a finding which has since been validated[37], whilst *CCR3* has been reported more recently[38]. The HLA-DRB1*0103 haplotype has previously been reported as associated with ulcerative colitis[39,40] and both Crohn's and ulcerative colitis[41] in candidate gene association studies. It is possible that the overlap in lead variants between these diseases is driven by specific tag variants flagging particular biological events affecting auto-immune traits of the digestive system.

Immune regulatory loci identified here may influence the susceptibility of infective or non-infective risk factors for mouth ulcers. A range of viral[42–44], bacterial[45] and other changes in the oral microbiome[46] have been suggested as acquired risk factors for mouth ulcers. It is possible that the genetic loci identified in this study relate to mouth ulcers through regulation of the host microbiome; a similar view has recently been proposed for both Behçet's and coeliac disease[47,48]. Hematinic status, especially deficiencies of folate, vitamin B12, ferritin or haemoglobin are thought to be risk factors for mouth ulcers[49] and these deficiencies might occur secondary to genetically determined inflammatory or immune states, such as pernicious anaemia or coeliac disease.

Loci identified in the present study may also have an effect at a tissue response level, where the cellular response to minor oral trauma is either proportionate and leads to resolution or disproportionate, leading to the clinical experience of mouth ulcers. One current view is that dysregulation of local cell-mediated response leads to an inappropriate focal accumulation of CD8+ T cell populations within the oral mucosa following minor triggers, leading to tissue damage and clinical manifestation as oral ulceration[44,50]. These upstream and downstream processes are not mutually exclusive and future research may wish to examine the effect of specific genetic loci across strata of potential risk factors to gain further insight into the aetiology.

A limitation of this study is that the presence or absence of mouth ulcers was inferred from questionnaire data rather than clinical examination. This is a necessary limitation as clinical oral examination data were not available and because the short duration and intermittent nature of mouth ulcers means they are often not visible on clinical examination even for affected individuals. In common with any questionnaire-derived data this may lead to some misclassification. For example, the phenotypes used in this study do not distinguish between RAS (which is in any case a clinical diagnosis) and other causes of mouth ulcers. For the most part we believe non-RAS ulceration (such as traumatic

ulceration) will be uncorrelated with genotype, and will therefore bias single SNP results and heritability estimates towards the null rather than generating false positive findings. Participants with Behçet's disease or ulcerative colitis may report mouth ulcers which are secondary to their underlying diagnosis which could lead to false positive findings, but these conditions are uncommon compared to RAS. There may also be participants who have a genetic predisposition to over-report or under-report their symptoms which would mean misclassification is correlated with genotype. As discussed above these non-specific associations across the genome are anticipated to have small effects and are therefore considered unlikely to influence the lead-variant results but may bias the heritability estimates away from the null.

Mouth ulcers become less common with age and there may be a genuine shift in aetiology with time. This was reflected in the different samples; the studies which include younger participants (ALSPAC and TW) had much higher mouth ulcer prevalence compared to the studies with older participants (UK Biobank and AG). A major motivation for the use of the look-up collections was to assess the effects of the lead genetic variants on more RAS-specific phenotypes and in studies with younger participants. However, a limitation of this strategy is the wide confidence intervals for effect estimates in each of the three smaller lookup cohorts, meaning that only variants with larger effects are expected to exclude the null and the heterogeneity across the three studies which precludes the use of meta-analysis.

The Open Targets analysis identified 52 drugs which might recapitulate the effects of naturally occurring genetic variation. Some of these agents are tumour necrosis factor targets and are licenced for immune-related diseases, such as rheumatoid arthritis (Infliximab[51]) while others such as Ustekinumab target IL12A and are licenced for several diseases (psoriasis[52], psoriatic arthritis[53] and Crohn's disease[54]), and are being repurposed for other immune diseases such as systemic lupus erythematosus[55]. This vignette may help illustrate how the availability of GWAS results for mouth ulcers could facilitate repurposing of existing drug interventions or the development of novel, specific interventions for mouth ulcers.

In conclusion, this GWAS of mouth ulcers identified multiple associated loci including a common variant near IL12A with large effects on risk of mouth ulcers. Follow-up analyses provide insight into the aetiology of this common ulcerative condition and prioritise topics for future basic and applied research.

## Methods

**Overview.** A GWAS for mouth ulcers was performed within UK Biobank. Variants passing a conventional threshold for genome-wide significance ($P \leq 5.0e{-}08$) were replicated in 23andMe. All variants which showed directional consistency across the two cohorts were further explored in three independent resources with more specific mouth ulcer phenotypes and genetic data; namely the ALSPAC, the QIMR Over 50s (Aged) study (AG) and the QIMR Melanocytic Naevi in Adolescent Twins (TW) study.

**Participants and phenotypes.** UK Biobank is a population-based health research resource consisting of ~500,000 people, aged between 38 and 73 years, who were recruited between the years 2006 and 2010 from across the UK[56]. Participants provided a range of information pertinent to adult and later life health outcomes via questionnaires, interviews, physical measurement and donating biological samples (data showcase available at www.ukbiobank.ac.uk)[57]. In the baseline questionnaire participants were asked to supply information about their oral health. The question stem was: Do you have any of the following? (You can select more than one answer). The possible answers included mouth ulcers and participants were prompted to answer this question thinking about the past year, if they pressed the help button. Participants who selected this answer were coded as cases. Participants who did not select this answer were coded as controls. Participants who chose the option for prefer not to answer, or did not complete the questionnaire session were coded as missing and not included in further analysis. UK Biobank received ethical approval from the North West Multi-centre Research Ethics Committee (REC reference for UK Biobank is 11/NW/0382).

23andMe Inc. is a personal genomics company that provides genotype and health-related information to customers[58]. Individuals included in the analyses provided informed consent and answered surveys online in accordance with the 23andMe human subjects protocol, which was reviewed and approved by Ethical and Independent Review Services, a private institutional review board (http://www.eandireview.com). Mouth ulcer cases were defined as those who answered yes to the question: Have you ever had a canker sore (an open sore on the soft tissue inside the mouth)? Those who answered no were considered controls, and those who responded don't know were not included in the analysis.

The ALSPAC longitudinal birth cohort recruited pregnant women living near Bristol, UK with an estimated delivery date between 1991 and 1992. There were 15,247 pregnancies resulting in 14,973 live births[59]. Follow up has included clinical assessment and questionnaires and is ongoing. Ethical approval for the study was obtained from the ALSPAC Ethics and Law Committee and the Local Research Ethics Committees. Informed consent for the use of data collected via questionnaires and clinics was obtained from participants following the recommendations of the ALSPAC Ethics and Law Committee at the time. Study children were asked to complete questionnaires about oral health at age 23.9 years. Participants were asked if they had ever had mouth ulcers (no/yes, but only once or twice/yes, on several occasions). An ever ulcers phenotype was generated with any answer of Yes,… used to define case status and No to define controls. Those who did not answer the question were set to missing. Study data were collected and managed using REDCap electronic data capture tools[60].

The QIMR Over 50s (AG) study was initiated in 1992–1993 to understand the role of genetics in healthy aging and age-related disease. AG recruited twin pairs from the Australian Twin Register who were over 50 years of age. This study was approved by the QIMR Berghofer Human Research Ethics Committee (HREC reference number P1204). Participants were asked whether they had mouth ulcers now/previously/both or never. This was used to derive case control status for ever or never having ulcers.

The QIMR melanocytic naevi in adolescent twins (TW) study was established in 1992 to investigate melanotic naevae. Twin pairs aged around 12 years of age were recruited from schools in Brisbane and the surrounding area in Queensland, Australia. This study was approved by the QIMR Berghofer Human Research Ethics Committee (HREC reference number P193). All twins and most parents donated blood for DNA extraction and completed a questionnaire. Children and mothers of nuclear families were asked independently about mouth ulcers. Mothers also answered a questionnaire on mouth ulcers on behalf of the entire family, with each family member given a score for ulcers which was scaled to allow comparison with the studies with binary ulcer phenotypes (never = 0, rarely = 1/3, sometimes = 2/3, frequently = 1). Pictures of RAS were used to help participants identify whether they had this specific type of oral ulceration. These ulcer questions were treated as a severity score preserving all four possible responses. Severity scores from both parents and children were combined into a mixed cohort.

Oral health phenotypic data collection in the ALSPAC study was conducted between November 2015 and September 2016, the AG study was conducted between 1992 and 1993 ($n = 1120$). Mouth ulcer data in the TW study were collected from 1992 until 2016 ($n = 2442$).

**Genotypes.** The UK Biobank genotype data (July 2017 release) contains 488,377 successfully genotyped samples. 49,979 individuals were genotyped using the UK BiLEVE array and 438,398 using the UK Biobank axiom array. Pre-imputation QC, phasing and imputation were completed[61]. In brief, prior to phasing, multiallelic variants or those with minor allele frequency ≤ 1% were removed. Phasing of genotype data was performed using a modified version of the SHAPEIT2 algorithm[62]. Genotype imputation was performed to a combined UK10K haplotype[63] and Haplotype reference consortium (HRC) reference panels using IMPUTE2 algorithms[64]. A further QC protocol was then applied at the Wellcome Trust Centre for Human Genetics prior to release (http://biobank.ctsu.ox.ac.uk/crystal/docs/genotyping_qc.pdf). The analyses presented here were restricted to autosomal variants using a graded filtering so that rarer genetic variants are required to have a higher imputation INFO score (Info > 0.3 for MAF > 3%; Info > 0.6 for MAF 1–3%; Info > 0.8 for MAF 0.5–1%; Info > 0.9 for MAF 0.1–0.5%). Monomorphic and uncommon variants with MAF < 0.1% were removed. In addition, all variants not included in the HRC site list were removed. Individuals with sex-mismatch (derived by comparing genetic sex and reported sex) or individuals with sex-chromosome aneuploidy were excluded from the analysis ($n = 814$). A $k$-means cluster analysis was performed with four clusters using the first four principal components provided by UK Biobank in the statistical software environment R[65]. The current analysis includes the largest cluster from this analysis ($n = 464,708$), of whom 461,106 individuals (47,100 with mouth ulcers) had non-missing phenotype data and were included in GWAS[66].

Sensitivity analyses of the HLA region used imputed HLA haplotypes provided by UK Biobank. As this analysis was performed using logistic regression rather than mixed models, analysis was restricted to individuals of white British ancestry. In addition, one individual from each pair of closely related (third degree or closer) individuals was removed until no related pairs remained. Following these exclusion criteria, 337,115 individuals had genotype data available, of which 336,038 had non-missing phenotype data and were included in logistic regression models.

In 23andMe, DNA extraction and genotyping were performed on saliva samples by CLIA-certified and CAP-accredited clinical laboratories of Laboratory Corporation of America. QC, imputation and genome-wide analysis were performed by 23andMe[67]. Briefly, samples were genotyped on a 23andMe custom genotyping array platform (Illumina HumanHap550+Bead chip V1 V2, OmniExpress+Bead chip V3, custom array V4) with a minimum call rate of 98.5%. Missing participant genotype date was imputed using the UK10K and 1000 genomes combined reference panel. Analysis was conducted on a maximal set of unrelated individuals.

Genetic data for the ALSPAC participants has been collected ($n > 10,000$). Genotyping was conducted on the Illumina HumanHap550 quad chip for children and Illumina human660W quad array for mothers. Prior to imputation, samples with >3% missingness, indeterminate heterozygosity, extreme autosomal heterozygosity or which clustered outside the CEU HapMap2 population using multidimensional scaling were removed. In addition, variants with minor allele frequency of <1%, call rate of <95% or violations of Hardy–Weinberg equilibrium ($P < 5e-07$) were removed. Samples and variants passing these QC measures were carried forward to a joint phasing stage, prior to imputation to the HRC reference panel using the Michigan imputation server (r1.1, 2017 release).

Genetic data for the participants of AG and TW with mouth ulcer data have been collected as part of a larger project by QIMR that comprises multiple waves of genotyping. In total, 3562 participants had phenotype and genotype data available ($n = 1120$ for AG, $n = 2442$ for TW). Most participants were genotyped on the Illumina Human610-Quadv1_B ($n = 1942$) or HumanCoreExome-12v1-0_C ($n = 1019$) arrays; genotyping on a small number of additional individuals was conducted on the Illumina HumanOmni25M-8v1-1_B ($n = 144$), 317 K ($n = 247$), HumanCNV370 ($n = 91$), Human660W-Quad_v1_C ($n = 8$), HumanOmniExpress-12v1-1_A ($n = 21$) and PsychArray-B ($n = 92$) platforms. Genotype data from all assays was jointly imputed using HRC reference panel (r1.1)[68–70].

**Statistical methods**. A GWAS was performed using a LMM implemented in BOLT-LMM (v2.3)[71,72] using an in-house GWAS pipeline[73]. A subset of 143,006 high quality variants were used to estimate and account for genetic relatedness and ancestry, allowing for the inclusion of closely and distantly related individuals in genetic analyses[74]. Age, sex and genotyping array were included as covariates in association testing. The Bayesian model was not predicted to provide a substantially better fit than a conventional LMM[72], so results are presented for the standard bolt_lmm_inf model. BOLT-LMM association statistics are on the linear scale. As such, test statistics (betas and their corresponding standard errors) were transformed to log odds ratios and their corresponding 95% confidence intervals on the liability scale using a Taylor transformation expansion series[75]. Other methods for transforming betas to odds ratios that take into account allele frequency have been suggested. Both methods for deriving transformed OR showed excellent concordance (Supplementary Figure 12) so only the values from Taylor transformation expansion series are reported. Genome wide significance was defined at $P < 5.0e-08$.

Associated loci that passed the genome-wide association threshold were clumped based on LD values using PLINK (version 1.9)[76] to identify approximately independent associated variants (options --clump-kb 10000 –clump-p1 5e-08 --clump-p2 1 --clump-r2 0.1), using an independent sample of HRC imputed genetic data to estimate LD. These variants were selected for replication in 23andMe.

In 23andMe, associations between the UK Biobank genome-wide associated variants and the mouth ulcer phenotype were assessed using linear regression assuming an additive model and using a 23andMe internally developed pipeline. Replication was considered successful if directionality was consistent across the two studies. Where replication was successful the estimates from UK Biobank and 23andMe were meta-analysed using the meta package in R[77].

Association between the replicated variants and the mouth ulcer phenotypes were assessed using a LMM with age and sex as covariates in the three lookup cohorts. In ALSPAC this was implemented in BOLT-LMM (v2.3)[71,72] as described above for UK Biobank. For AG and TW this was implemented in RareMetalWorker[78]. As with the GWAS association statistics, the results were transformed using a Taylor transformation expansion series to express log odds ratios on the liability scale. Variant lookups were in additional collections with better phenotypic data. These lookups aimed to provide conducted to provide complementary evidence for the effects sizes of the lead variants from independent sources of data. Insufficient power in individual studies and heterogeneity in phenotype across studies, which precludes meta-analysis, mean they were not used as further replication sets. Therefore, all variants identified in UK Biobank and followed up in 23andMe were taken through to in-silico functional analyses to avoid incorrectly disregarding potentially causal variants.

**Tests for polygenicity bias and heritability**. Univariate LD score regression was performed using the LDSC software package (v1.0.0, April 2017 release)[23] to quantify the relative contribution of polygenicity and bias to inflation in single variant test statistics from UKBB. We used LDSC to estimate heritability under the infinitesimal model assumption and obtained contrasting estimates from the HESS method which may be more robust under certain genetic architectures[24].

**HLA haplotype analysis**. Analysis of the HLA region in UK Biobank used 362 imputed haplotypes (provided by UK Biobank). Unadjusted analysis was performed using logistic regression and an additive model for each haplotype which used haplotype dosage to account for uncertainty in haplotype imputation. Seventeen haplotypes were not observed within the final sample, whilst 345 were present in at least one individual and produced test statistics. Of these 345 haplotypes, 24 passed a Bonferroni-corrected $P$-value threshold ($P < 1.5e-04$) and were examined further in a fully adjusted model which included adjustment for age, sex, 40 genetic principal components and genotyping array.

**In-silico functional analysis**. Tests for association between ulcers and predicted gene expression were performed using S-PrediXcan[26], which uses predictive models to impute transcript expression levels in specific tissues and then uses full GWAS summary statistics to test for associations between these predicted expression levels and phenotype. A threshold was applied on predicted performance (a measure of ability to accurately infer transcription levels at a given locus in a specific tissue) with a cut-off at 0.1. After filtering, all remaining predicted expression levels were tested for association with mouth ulcers. S-MulTiXcan[28] was then applied to these tissue-specific estimates, which harnesses information from the sharing of expression quantitative trait loci (eQTL) across multiple tissues, to increase power to detect the effects of gene expression on the phenotype of interest. A Bonferroni correction for multiple testing was applied which takes into the correlation between tissues and equates to 0.05 divided by the number of genes with a prediction model in at least one tissue ($P < 1.9e-06$). Analysis was performed using the MetaXcan standalone package, which includes the S-PrediXcan and S-MulTiXcan methods.

Gene prioritisation, gene set enrichment and tissue enrichment analysis were performed using the DEPICT software package[79]. DEPICT anticipates that association signals for a biologically causal gene will co-localise with genes encoding other members of a co-regulated gene network elsewhere in the genome. By using publicly available gene expression data to define co-regulated gene networks, DEPICT can take account of association patterns elsewhere in the genome to nominate biologically plausible candidate genes at a statistically associated locus. DEPICT analysis was performed using variants that passed the genome-wide threshold ($P < 5e-8$) after clumping and default settings in the standalone java software.

Enrichment in regulatory motifs was assessed using the non-parametric enrichment analysis package GARFIELD[25]. Garfield performs greedy LD pruning of GWAS summary statistics using reference data from the UK10K project. LD-tagged genomic regions are then annotated with information on 1005 regulatory features identified in ENCODE, GENCODE and Roadmap epigenomics projects. These features include genic annotations, chromatin states, histone modifications, DNase1 hypersensitive sites and transcription factor-binding sites in a range of cell lines. GARFIELD tests whether regulatory information is present in trait-associated loci more frequency than expected by chance by performing adaptive permutations at various significance thresholds ($p = 10^{-1}, 10^{-2},…, 10^{-8}$) but requires full GWAS data. Trait-associated loci are matched with appropriate null loci using many features, such as LD information from the ENCODE and ROADMAP projects. Analysis was conducted using default settings in standalone GARFIELD (v1.0) software.

**Tests for genetic correlation**. Genetic correlation between mouth ulcers and other traits was assessed using bivariate LD score regression implemented in the LD Hub platform[29]. For traits that passed the Bonferroni-corrected threshold for association the rho-HESS approach[80] was applied to estimate local correlation between mouth ulcers and these phenotypes.

**PRS analysis**. PRSs were generated in plink v1.9 at a range of $P$-value thresholds between 1 and $5e-08$. PRS were standardised using a mean of 0 and a standard deviation of 1. Association between PRS and mouth ulcers was assessed using LMMs in GCTA[30], including a genetic relatedness matrix to account for family structure and relatedness in QIMR.

**Drug repurposing**. The Open Targets database for pharmacological interventions (https://www.targetvalidation.org) was used to assess whether any of the ulcer-associated genes might represent targets for drug repurposing. The S-PrediXcan analysis was used to select gene transcripts.

## Data availability
All phenotypes and genotypes used in UK Biobank are available to bona fide researchers through a process described at ukbiobank.ac.uk. The ALSPAC study website contains details of all the data that are available through a fully searchable data dictionary (www.bristol.ac.uk/alspac/researchers/our-data). The data access procedures for ALSPAC data are described in full online (http://www.bristol.ac.uk/alspac/researchers/access/). Researchers interested in using QIMR data can contact Nick Martin (Nick.Martin@qimrberghofer.edu.au). All software packages referred to in the methods are available online. The summary statistics of the UK Biobank GWAS have been deposited at data.bris [https://doi.org/10.5523/bris.459eyiulzf9y25yh6nsf550y4] and the summary statistics for the 97 lead variants from 23andMe, inc. and the meta-analysis of these with the UK Biobank variants can be found in Supplementary Data 1.

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

## Acknowledgements

S.H. and T.D. work in a unit that receives funding from the University of Bristol and the UK Medical Research Council (Grant ref: MC_UU_12013/3). T.D. receives support from Wellcome (Grant ref: 201268/Z/16/Z). S.H. receives support from Wellcome (Grant ref: 201237/Z/16/Z). N.J.T. is a Wellcome Trust Investigator (202802/Z/16/Z), is the PI of the Avon Longitudinal Study of Parents and Children (MRC & WT 102215/2/13/2), is supported by the University of Bristol NIHR Biomedical Research Centre (BRC-1215-20011), the MRC Integrative Epidemiology Unit (MC_UU_12013/3) and works within the CRUK Integrative Cancer Epidemiology Programme (C18281/A19169). The UK Biobank was established by the Wellcome Trust, Medical Research Council, Department of Health, Scottish Government and the Northwest Regional Development Agency. It has also received funding from the Welsh Assembly Government, British Heart Foundation and Diabetes UK. This research has been conducted using the UK Biobank Resource under Application Number '40644'. The UK Medical Research Council and Wellcome (Grant ref: 102215/2/13/2) and the University of Bristol provide core support for ALSPAC. This publication is the work of the authors and Nicholas Timpson will serve as guarantor for the contents of this paper. A comprehensive list of grants funding (PDF, 459KB) is available on the ALSPAC website. This research was specifically funded by the University of Bristol. GWAS data was generated by Sample Logistics and Genotyping Facilities at Wellcome Sanger Institute and LabCorp (Laboratory Corporation of America) using support from 23andMe. We are extremely grateful to all the families who took part in this study, the midwives for their help in recruiting them, and the whole ALSPAC team, which includes interviewers, computer and laboratory technicians, clerical workers, research scientists, volunteers, managers, receptionists and nurses. We thank the research participants and employees of 23andMe for making this work possible, especially the members of the 23andMe Research Team. Funding for genotyping in QIMR was provided by the Australian National Health and Medical Research Council (241944, 339462, 389927, 389875, 389891, 389892, 389938, 442915, 442981, 496739, 552485, 552498), the Australian Research Council (A7960034, A79906588, A79801419, DP0770096, DP0212016, DP0343921), the FP-5 GenomEUtwin Project (QLG2-CT-2002-01254), the U.S. National Institutes of Health (NIH grants AA07535, AA10248, AA13320, AA13321, AA13326, AA14041, DA12854, MH66206), and Mr. George Landers (the Over 50s [AG] study). A portion of the genotyping on which the QIMR study was based (Illumina 370K scans) was carried out at the Center for Inherited Disease Research, Baltimore (CIDR), through an access award to the authors' late colleague Dr. Richard Todd (Psychiatry, Washington University School of Medicine, St. Louis). L.C.-C. is supported by a QIMR Berghofer Fellowship. S.E.M. is supported by an Australian National Health and Medical Research Council Fellowship (APP1103623). The funders had no role in study design, data collection and analysis, decision to publish, or preparation of the manuscript. Researchers interested in using QIMR data can contact Nick Martin (Nick.Martin@qimrberghofer.edu.au).

## Author contributions

S.H. and T.D. conducted the ALSPAC data collection, UK Biobank and ALSPAC analysis, conducted the meta-analysis and in-silico functional analyses, interpreted the results and wrote the manuscript. N.J.T. supervised the analysis, interpreted the results and edited the manuscript. R.M., B.E. and L.P. created the infrastructure to enable the UK Biobank analysis within the MRC IEU and provided comments on the manuscript. P.A.L., L.C.-C., S.E.M. and S.G. analysed the TW and AG datasets and provided comments on the manuscript. S.J.T. and N.G.M. collected the TW and AG data and edited the manuscript. P.W.F. interpreted the results and provided comments on the manuscript. J.F.S. contributed replication in the 23andMe dataset and was supervised by J.Y.T. The 23andMe Research Team created the infrastructure to enable data collection and analysis.

## Additional information

**Competing interests:** J.F.S., J.Y.T. and members of the 23andMe Research Team are employees of 23andMe, Inc. and hold stock or stock options in 23andMe. The remaining authors declare no competing interests.

**23andMe Research Team**

Michelle Agee[4], Babak Alipanahi[4], Adam Auton[4], Robert K. Bell[4], Katarzyna Bryc[4], Sarah L. Elson[4], Pierre Fontanillas[4], Nicholas A. Furlotte[4], Barry Hicks[4], David A. Hinds[4], Karen E. Huber[4], Ethan M. Jewett[4], Yunxuan Jiang[4], Aaron Kleinman[4], Keng-Han Lin[4], Nadia K. Litterman[4], Jennifer C. McCeight[4], Matthew H. McIntyre[4], Kimberly F. McManus[4], Joanna L. Mountain[4], Elizabeth S. Noblin[4], Carrie A.M. Northover[4], Steven J. Pitts[4], G. David Poznik[4], Janie F. Shelton[4], Suyash Shringarpure[4], Chao Tian[4], Vladimir Vacic[4], Xin Wang[4] & Catherine H. Wilson[4]

