## [Peer Review File · Nature Communications]

Reviewer #1 (Remarks to the Author):

The authors describe a large genome-wide association study with the aim to identify genetic factors that contribute to risk for a common phenotype, predilection for mouth ulcers. The discovery collection was from the UK Biobank, in which about 10% of participants (cases) self-reported having mouth ulcers within the last year. After accounting for LD, the authors identified 97 approximately independent lead variants with genome-wide significance. In the replication collection from 23andMe, more than 72% of participants self-reported “ever having canker sores” (cases). Despite this grossly different case frequency, the authors report that all the 97 variants showed directional consistency and comparable effect size in the 23andMe collection. The authors also examined these 97 variants in three smaller collections with more recurrent aphthous ulcer-specific phenotype and many of the associations replicated in one or more of the collections, although the case frequencies in these collections were also very different (74.0%, 18.7%, and 86.8%). In addition, the authors used informatic tools to prioritize the genes that could be influenced by the ulcer-associated variants. They also examined the locations of the variants relative to known genes and regulatory elements, the biological pathways that the associated variants may influence, and the tissue/cell types associated with the identified genes’ expression. They also used the genotype data to estimate gene expression and test the association of predicted gene expression with ulcer risk. They also performed HLA haplotype analyses and tested whether a polygenic risk score developed from the UK Biobank data could explain the variance in mouth ulcer phenotype of individuals in two of the smaller collections. Lastly, the authors attempted to identify genetic correlations of the mouth ulcer phenotype with other traits, although this is perhaps the least convincing part of the report.

The genetic findings of this study are remarkably robust despite vastly different self-identified case status frequencies among the collections studied. Therefore, the conclusions about the role played by immune regulatory loci can also be considered quite strong. This study demonstrates the utility of large studies with self-reported phenotypes that might have previously been considered too imprecise to be informative. Some specific comments follow.

1. The results section states that the phenotype used for the UK Biobank collection was self-reported “having mouth ulcers within the last year”. The methods section states that the phenotype was a “yes” answer to the question “Do you have any of the following....with one selection “mouth ulcers”. Please clarify in the methods whether “within the last year” was specified.
2. Line 142: should refer to Figure 2c instead of Figure 2b.
3. Clarify whether the DEPICT, GARFIELD, and PrediXcan analyses were performed with full GWAS data or with a subset based on significance level or LD-pruning, for example.
4. In the polygenic risk score analysis, the numbers in lines 224 and 225 do not seem to match those in supplementary Tables 7 and 8. Furthermore, there is no discussion of the low percentage of variation in mouth ulcer severity or case status explained by these risk scores.
5. The test for genetic correlation gave unexpected results (neuroticism and depressive symptoms), considering that the authors identified immune regulatory loci associated with oral ulcers. This disparity is hard to reconcile and therefore requires a higher level for proof, for example replication with another study or another tool and also requires a description and explanation of the loci driving the correlation. Alternatively, this correlation analysis could be omitted.

6. HLA haplotype analysis (lines 213-219). Report the DRB1*0103 haplotype frequencies in the cases and in the controls or the frequency of cases and the frequency of controls with the haplotype (about double the haplotype frequency).
7. The discussion of oral ulcer genetic variants in common with those identified in Behcet's disease (BD) should be expanded to include variants described by Takeuchi et al, Nature Genetics 49: 438, 2017, which reports seven variants with strong LD with one of the 97 oral ulcer variants, all with consistent direction of effect (see below).
8. Lines 307-308: states "increased expression of CCR2 in whole blood is associated with increased odds of developing mouth ulcers..." Please mark the CCR2 gene on Figure 3.
9. Regarding the application of GWAS results to suggest drug interventions, the authors should discuss promising studies of Apremilast (a PDE4 inhibitor) for prevention of oral ulcers in Behcet's disease (for example, Hatemi, et al NEJM 372:1510, 2015), in light of the oral ulcer non-coding PDE4D variant identified in this study, rs4235479. It would also be informative to mark this gene (PDE4D) on Figure 3.

Gene	BD-variant	Oral ulcer-variant	r2 Eur*	D' Eur*
IL10	rs1581110	rs1800871	0.89	0.98
STAT4	rs7574070	rs11684030	0.93	0.99
IL12A	rs17753641	rs76830965	1	1
RIPK2	rs2230801	rs11989430	0.55	0.94
IRF8	rs7203487	rs7193275	0.98	1
IRF8	rs142105922	rs11649485	0.85	0.94
CEBPB-PTPN1	rs913678	rs913678	1	1

*LD data from Haploreg

RIPK2	**rs10094579	rs11989430	0.87	0.97
-------	--------------	------------	------	------

**Lead RIPK2 marker from BD discovery collection

Reviewer #2 (Remarks to the Author):

The authors perform a case-control association study to investigate the genetic susceptibility to recurrent aphthous stomatitis (RAS). They use genetic data from UK Biobank and identify 97 independent loci exceeding genome-wide significance which are then validated for directional effect in data from 23andMe. Given the non-specific phenotype definition for mouth ulcers in these two datasets the authors then attempt to validate these in a series of datasets where they are confident

of the RAS phenotype and replicate 24 of these loci. The identified loci are then used to infer biological insight using a range of bioinformatic software.

Major findings:

1. 97 genome-wide significant loci. 24 of which validate in clinically relevant datasets
2. An accurate estimate of heritability
3. Bioinformatic analysis reveals involvement of immune genes and immune relevant cell types

The paper is well written and my comments are below:

- My main concern is the use of consistent directional effects as a method for determining validation. This is a particular problem when trying to interpret the results from the three clinical validation cohorts. Many of the SNPs that show consistency of effect direction across the three groups (i.e. considered to be replicated) are not statistically significant in any of the groups and the confidence intervals of many include 1. I appreciate the authors are concerned about the power of the clinical validation cohorts; a meta-analysis of the three clinical cohorts would capture the consistency of effects and improve power. Few of the 24 claimed validated loci appear to be associated with RAS in the conventional sense.
- The results for the 24 SNPs validated by the three clinical dataset are difficult to extract from supplementary table 1 and it would benefit the reader if they were flagged or presented in a separate table.
- The heritability estimates are calculated in UK Biobank data which has a non-specific phenotype definition. The authors could comment on the impact that this definition has upon the interpretation of the estimate.
- The bioinformatic analysis was conducted using the 97 loci identified using a non-specific phenotype definition. Have the authors performed any sensitivity analysis just using the 24 loci that are considered to be validated in the more clinically relevant datasets
- In the discussion the authors introduce results from drug repositioning analysis. This analysis should be included in the methods/results section similar to the other analyses performed.

Title: Genome wide analysis for mouth ulcers identifies associations at immune regulatory loci.

Authors. Tom Dudding^{1,2#}, Simon Haworth^{1,2#}, Penelope A. Lind³, J. Fah Sathirapongsasuti⁴, the 23andMe Research Team⁴, Joyce Y. Tung⁴, Ruth Mitchell¹, Lucía Colodro-Conde³, Sarah E. Medland³, Scott Gordon⁵, Benjamin Elsworth¹, Philip Haycock¹, Lavinia Paternoster¹, Paul W Franks^{6,7,8}, Steven J Thomas², Nicholas G. Martin⁵, Nicholas J Timpson^{1*}

These authors contributed equally

* Corresponding Author

Please find below responses to all reviewer points. Reviewer comments are in italics and responses follow, altered text is shown in **bold font**. We have tried to include the changes in this response where we can but changes can also be seen as track changes on the marked copy of the manuscript.

REVIEWER 1:

1.1. The results section states that the phenotype used for the UK Biobank collection was self-reported "having mouth ulcers within the last year". The methods section states that the phenotype was a "yes" answer to the question "Do you have any of the following....with one selection "mouth ulcers". Please clarify in the methods whether "within the last year" was specified.

We thank the reviewer for pointing out the wording of the ulcer phenotype was not clear for UK Biobank. The "within the last year" timeframe was specified if the participants clicked on the help button. A sentence has been added to the methods to clarify this.

Methods:

"The possible answers included "Mouth ulcers" and participants were prompted to "Answer this question thinking about the past year" if they pressed the help button."

1.2. Line 142: should refer to Figure 2c instead of Figure 2b.

We have amended this figure numbering in the manuscript.

1.3. Clarify whether the DEPICT, GARFIELD, and PrediXcan analyses were performed with full GWAS data or with a subset based on significance level or LD-pruning, for example.

For the in-silico analyses, DEPICT uses a genome-wide threshold ($p < 5e-8$) after clumping while PrediXcan requires full GWAS data. GARFIELD iterates through a series of significance levels (10^{-1} , 10^{-2} , ..., 10^{-8}) but full GWAS data is required as a genome-wide distribution of p-values with a given trait is used to calculate the level of enrichment of regulatory or functional annotations. These thresholds have been clarified in the manuscript as detailed below:

DEPICT:

Results

"...in DEPICT was performed to nominate plausible biologically causal genes by identifying genes in different statistically associated loci (**$p < 5e-8$ after clumping**) that have similar ..."

Methods

"DEPICT analysis was performed using **variants that passed the genome-wide threshold ($p < 5e-8$) after clumping** and default settings in the standalone java software."

GARFIELD:

Methods

“GARFIELD tests whether regulatory information is present in trait-associated loci more frequency than expected by chance by performing adaptive **permutations at various significance thresholds ($p = 10^{-1}, 10^{-2}, \dots, 10^{-8}$) but requires full GWAS data. Trait-associated loci are matched** with appropriate null loci using many features such as LD information from the ENCODE and ROADMAP projects.”

PrediXcan:

Results

“PrediXcan,²⁶ which uses predictive models to impute transcript expression levels trained in 48 gene-tissue expression project (GTEx)²⁷ tissues and then **uses full GWAS summary statistics to test for association between these predicted expression levels and phenotype.**”

Methods

“Tests for association between ulcers and predicted gene expression were performed using S-PrediXcan,²⁶ which uses predictive models to impute transcript expression levels in specific tissues and then **uses full GWAS summary statistics to test for associations between these predicted expression levels and phenotype.**”

1.4. In the polygenic risk score analysis, the numbers in lines 224 and 225 do not seem to match those in supplementary Tables 7 and 8. Furthermore, there is no discussion of the low percentage of variation in mouth ulcer severity or case status explained by these risk scores.

We thank the reviewer for pointing out the numbers not matching those in the tables. We have gone back to the original results tables and updated the numbers ensuring both the manuscript and supplementary tables are correct.

As the reviewer points out, the polygenic risk scores explain only a small amount of variance in the ulcer phenotype in the QIMR datasets. We have added a line in the discussion suggesting the use of the genetic variants identified in this work as a predictive tool is unlikely to be clinically useful.

“PRSs, generated using variants selected with a range of significance thresholds, explained only a small amount of variance in the ulcer phenotypes. At present, these PRSs are unlikely to be a clinically useful tool in predicting which patients are at risk of developing mouth ulcers or in predicting severity of mouth ulcers”.

1.5. The test for genetic correlation gave unexpected results (neuroticism and depressive symptoms), considering that the authors identified immune regulatory loci associated with oral ulcers. This disparity is hard to reconcile and therefore requires a higher level for proof, for example replication with another study or another tool and also requires a description and explanation of the loci driving the correlation. Alternatively, this correlation analysis could be omitted.

We would like to keep a distinction between cross-trait comparison of lead associated single variants and genome-wide cross trait comparisons. These are complementary analyses intending to explore different aspects of the mouth ulcer phenotype, and we do not find it surprising that the results are somewhat different.

The single variant lookup only considered lead signals passing genome-wide significance in the UK Biobank GWAS. This procedure selects variants with medium to large effect estimates, which may flag specific biological processes relevant to ulcers. In this case there is evidence for enrichment in loci thought to be involved in the regulation of inflammatory profile. Furthermore, the single-variant cross trait lookup suggests that several of these specific variants are relevant for other inflammatory or autoimmune diseases.

In contrast to this, subtle effects on reporting (such as altered genetic propensity to notice having ulcers) will have effects which are too small to pick up at a single variant level but may be noticeable when aggregated across all common genetic variants as they can occur at any locus regardless of whether this is an inflammatory-regulatory region or not. This may explain the genetic correlation of ulcers with neuroticism and depression.

One interpretation explaining both the single variant and genetic correlation analysis is that at an aggregate level (i.e. incorporating all common genetic variation in the genome), mouth ulceration is a distinct trait from other auto-immune traits (i.e. a polygenic trait affected by many loci indirectly), but that specific variants are present which flag auto-immune traits in this architecture. Therefore, genome-wide correlation with neuroticism and depression can be explained by a genetic liability for personality traits to influence participant awareness or reporting of mouth ulcers, whilst specific biological pathways contribute and are seen in locus specific main effects.

If correct, this explanation gives a testable hypothesis that genetic correlation between mouth ulcers and personality traits should show distributed patterns of genetic overlap across the genome, without peaks coinciding with peaks in local heritability of mouth ulcers. To examine this further we used the rho-HESS approach to estimate local genetic correlation between mouth ulcers and these two phenotypes.¹ This approach estimates genetic correlation from genome-wide data by partitioning SNPs into approximately 1700 loci, estimating genetic covariance explained by each of these loci, and finally estimating genome-wide genetic correlation as the total of all local estimates.

We have been able to suggest that genetic correlation between mouth ulcers and both personality traits is distributed across the genome, without peaks in genetic correlation corresponding to peaks in local heritability of either mouth ulcers or these personality traits (**Review figures 1 and 2 at the end of this document**).

The estimates from rho-HESS gave consistent interpretation with r_g values from LDSR. For mouth ulcers versus neuroticism, the genome-wide genetic correlation estimate was $r_g = 0.18$, $P = 8.43e-107$ (LDSR: $r_g = 0.23$, $P = 1.80e-08$). For mouth ulcers versus depressive symptoms the genome-wide genetic correlation estimate was $r_g = 0.33$, $P = 4.16e-21$ (LDSR: $r_g = 0.24$, $P = 5.73e-07$).

We have altered the manuscript to reflect these views and have discussed it further below:

Methods:

“For traits that passed the Bonferroni corrected threshold for association the rho-HESS approach⁷⁹ was applied to estimate local correlation between mouth ulcers and these phenotypes.”

Results:

“For neuroticism and depressive symptoms, the genetic correlation was further examined using the rho-HESS approach which estimates local genetic correlation between mouth ulcers and these traits. At an aggregate level (i.e. incorporating all common genetic variation in the genome), the r_g estimates from rho-HESS gave consistent interpretation with those from LDSR (neuroticism: $r_g = 0.18$, $P = 8.43e-107$; depressive symptoms: $r_g = 0.33$, $P = 4.16e-21$). Additionally, it shows that genetic correlation between mouth ulcers and these traits is evenly distributed across the genome, without peaks in genetic correlation corresponding to peaks in local heritability of either mouth ulcers or these two traits (Supplementary tables 6 and 7).”

Discussion:

“It is possible that the overlap in lead variants between these diseases is driven by specific tag variants flagging particular biological events affecting auto-immune traits of the digestive system.”

“The remainder of the heritability is likely driven by indirect effects of large numbers of variants with modest effects on the odds of developing mouth ulcers. Given these genetic variants, which act indirectly, likely also contribute to a wide range of distal health related phenotypes, it is perhaps unsurprising that neuroticism and depressive symptoms show detectable genetic correlation with mouth ulcers in both the LDSR and rho-HESS analyses. The view that this correlation is driven by non-specific associations across the genome, rather than effects at lead-associated variants, is supported by the location specific genetic correlation performed in rho-HESS.”

*1.6. HLA haplotype analysis (lines 213-219). Report the DRB1*0103 haplotype frequencies in the cases and in the controls or the frequency of cases and the frequency of controls with the haplotype (about double the haplotype frequency).*

We have altered the manuscript to report the DRB1*0103 haplotype frequency in both controls and cases. Additionally, we have added frequency in cases and frequency in controls columns to Supplementary Table 8 so that this information is available for all haplotypes passing the Bonferroni correction.

“The most robust finding was DRB1*0103, an uncommon haplotype (frequency in controls = 0.017, frequency in cases = 0.022) which...”

1.7. The discussion of oral ulcer genetic variants in common with those identified in Behcet’s disease (BD) should be expanded to include variants described by Takeuchi et al, Nature Genetics 49: 438, 2017, which reports seven variants with strong LD with one of the 97 oral ulcer variants, all with consistent direction of effect (see below).

We are very grateful to the reviewer for pointing out this relevant reference and for providing the LD information from Haploreg. These additional variants clearly show further overlap in the genetic signals of Behcet’s disease and non-specific mouth ulcers. We have added to the paragraph in the discussion to reflect this additional overlap.

“Genetic variants at *IL12A*, *IL10*, *STAT4*, *RIPK2*, *IRF8* and *CEBPB-PTPN1* which have been reported in previous GWAS for Behçet’s disease,³²⁻³⁴ are in high LD and have consistent effect direction with those reported here. This raises the possibility of a similar mechanism leading to clinical presentation with mouth ulcers in both conditions.”

1.8. Lines 307-308: states “increased expression of CCR2 in whole blood is associated with increased odds of developing mouth ulcers...” Please mark the CCR2 gene on Figure 3.

We have added the CCR2 label to Figure 3.

1.9. Regarding the application of GWAS results to suggest drug interventions, the authors should discuss promising studies of Apremilast (a PDE4 inhibitor) for prevention of oral ulcers in Behcet’s disease (for example, Hatemi, et al NEJM 372:1510, 2015), in light of the oral ulcer non-coding PDE4D variant identified in this study, rs4235479. It would also be informative to mark this gene (PDE4D) on Figure 3.

We thank the reviewer for drawing out attention to the studies of Apremilast in Behcet's disease which are potentially very relevant to this work. However, despite the strong association of rs4235479 with mouth ulcers in this study, the PrediXcan analyses did not predict expression of *PDE4D* would be strongly associated with mouth ulcers (strongest evidence of association across all tissues tested, $p = 0.19$). For this reason we have not labelled this gene in Figure 3.

However in response to reviewer 2 (point 5), we have expanded our investigation of drug repurposing in the light of the results of our study. In these changes we point out drugs which have the potential to be repurposed for use against mouth ulcers. These include other anti-TNF drugs such as Infliximab. Please see point 2.5 below.

REVIEWER 2:

2.1 My main concern is the use of consistent directional effects as a method for determining validation. This is a particular problem when trying to interpret the results from the three clinical validation cohorts. Many of the SNPs that show consistency of effect direction across the three groups (i.e. considered to be replicated) are not statistically significant in any of the groups and the confidence intervals of many include 1. I appreciate the authors are concerned about the power of the clinical validation cohorts; a meta-analysis of the three clinical cohorts would capture the consistency of effects and improve power. Few of the 24 claimed validated loci appear to be associated with RAS in the conventional sense.

We thank the reviewer for their comments regarding the strategy for determining replication in the smaller clinical cohorts. As mentioned in your comment, these studies are underpowered to identify effects of those seen for the majority of the lead variants in the conventional sense, we therefore feel it is inappropriate to use these effects as an additional round of replication. Our rationale for using these smaller cohorts was therefore to provide additional information on the effect size estimates in these better phenotyped populations, rather than to assess the validity of the variants with robust evidence from UK Biobank and 23andMe. The reviewer has brought to our attention that this was not clear in the manuscript and we have made changes to reflect this.

We considered the suggestion to perform a meta-analysis of the three clinical cohorts. There are marked differences between these cohorts in terms of age which might mask age-specific effect estimates or result in heterogeneity in the meta-analysis. We therefore feel that combining results in meta-analysis is less useful to the reader than providing study-specific estimates, which is the strategy we have adopted here.

Abstract:

"Effect estimates of these variants were also obtained in three independent cohorts with more specific phenotyping and specific study characteristics"

Methods:

"Variant lookups were conducted in additional collections with better phenotypic data. These lookups aimed to provide complementary evidence for the effects sizes of the lead variants from independent sources of data. Insufficient power in individual studies and heterogeneity in phenotype across studies, which precludes meta-analysis, mean they were not used as replication sets. Therefore, all variants identified in UK Biobank and followed-up in 23andMe were taken through to in-silico functional analyses to avoid incorrectly disregarding potentially causal variants."

Results:

“Three smaller samples **were used to assess the effect sizes of the 97 lead variants with** more RAS-specific phenotypes.”

“**Complementary evidence for a** protective effect of the C allele was **seen** in all three lookup cohorts”

Discussion:

“**Effect estimates** in samples with RAS-specific phenotypes **provide complementary evidence** that these variants are associated with this specific type of oral ulceration.”

“**Complementary** evidence of these variants having true causal associations with RAS is provided by the consistent effect directions seen for many of them in the three independent look-up cohorts with more specific phenotypes.”

“A major motivation for the use of the look-up collections was **to assess the effects of the lead genetic variants on more RAS-specific phenotypes and in studies with younger participants**. However, a limitation of this strategy is the wide confidence intervals for effect estimates in each of the three smaller lookup cohorts, meaning that only variants with larger effects are expected to exclude the null **and the heterogeneity across the three studies which precludes the use of meta-analysis.**”

2.2 The results for the 24 SNPs validated by the three clinical dataset are difficult to extract from supplementary table 1 and it would benefit the reader if they were flagged or presented in a separate table.

To make the direction of effect across studies easier to identify for the reader an additional column has been added to Supplementary Table 1 using +/- to indicate effect direction across the 5 studies. Where the studies show a consistent direction of effect, the row has been changed to bold typeface to allow the reader to easily identify these variants.

2.3 The heritability estimates are calculated in UK Biobank data which has a non-specific phenotype definition. The authors could comment on the impact that this definition has upon the interpretation of the estimate.

In the GWAS analysis we estimate the genetic effect of a single variant on reporting having ulcers. In general, under or over-reporting of questionnaire data would be expected to introduce noise rather than bias, which would result in under-estimation of effect sizes for causal SNPs and false negatives² rather than false positives. As the heritability estimates in both LDSR and HESS are derived from test statistics, we envisage that these would be biased towards 0 in the presence of misclassification, consistent with the interpretation provided in the manuscript that the SNP based heritability is an estimate of the ‘lower bound’ of heritability.

“Although **the estimate of heritability from this study is likely an under-estimate and only provides the lower bound, it is substantially** less than the heritability previously estimated in twin studies.”

However, we can also imagine situations in which misclassification is correlated with genotype, for example participants who are genetically predisposed to be more aware of their symptoms might systematically over-report symptoms, and this may be part of the explanation for the genome-wide correlation as discussed in response 1.5. This is now mentioned in the discussion section:

“For the most part we believe non-RAS ulceration (such as traumatic ulceration) will be uncorrelated with genotype and will therefore bias **single SNP results and heritability estimates** towards the null rather than generating false positive findings. Participants with Behçet’s disease or ulcerative colitis may report mouth ulcers which are secondary to their underlying diagnosis which could lead to false positive findings, but these conditions are uncommon compared to RAS. **There may also be participants who have a genetic predisposition to over- or under-report their symptoms which would mean misclassification is correlated with genotype. As discussed above these non-specific associations across the genome are anticipated to have small effects and are therefore considered unlikely to influence the lead-variant results but may bias the heritability estimates away from the null.**”

2.4 The bioinformatic analysis was conducted using the 97 loci identified using a non-specific phenotype definition. Have the authors performed any sensitivity analysis just using the 24 loci that are considered to be validated in the more clinically relevant datasets

For the GARFIELD and PrediXcan bioinformatic analyses, the software uses genome-wide summary statistics and therefore limiting the analysis to only the 24 variants with consistent effect directions is not possible.

As discussed in 2.1, we do not feel that the contributory evidence from the smaller studies is powerful enough to exclude any lead variants identified in the genome-wide analysis. However, we thank the reviewer for pointing out the clear potential of assessing the DEPICT analysis using only the 24 variants with consistent effect direction. Several of these 24 variants cluster around the HLA region (which is excluded by DEPICT) and other of these variants represent independent signals of association within the same genomic region. This means there are only 10 distinct genomic regions in this sensitivity analysis, which is the absolute minimum required by DEPICT to assess for enrichment. We have included results from this analysis here (**Review Table 1,2 and 3**), however due to the limited number of genomic regions, power for these analyses is limited. We have retained only the full DEPICT analysis in the manuscript.

2.5 In the discussion the authors introduce results from drug repositioning analysis. This analysis should be included in the methods/results section similar to the other analyses performed.

In the initial manuscript we included the drug repurposing in the discussion as a brief vignette. We agree with reviewer 2 that the additional of this as a formal method in this work would mean the analysis is more meaningful to the reader. We have therefore performed the analysis again, added a section to the results and describing the analysis in the methods.

Methods:

“Drug repurposing

The Open Targets database for pharmacological interventions

(<https://www.targetvalidation.org>) was used to assess whether any of the ulcer-associated genes might represent targets for drug repurposing. The S-PrediXcan analysis was used to select gene transcripts.”

Results:

“Drug repurposing

To assess whether associated loci might represent targets for repurposed drug interventions we examined the Open Targets database for pharmacological interventions which might recapitulate the effects of naturally occurring genetic variation. Of the 244 gene transcripts

that passed Bonferroni correction in S-PrediXcan, 27 were not recognised by the platform. As the platform limits the number of genes to 200, the 17 with the weakest evidence for association in S-PrediXcan were not included in the model. 52 drugs were identified as potential targets (Supplementary table 9). Fourteen of these are in phase IV trials including Ustekinumab, an antibody against the IL12 protein, encoded by the *IL12A* gene.”

Discussion:

“The Open Targets analysis identified 52 drugs which might recapitulate the effects of naturally occurring genetic variation. Some of these agents are Tumor Necrosis Factor targets and are licenced for immune related diseases such as rheumatoid arthritis (Infliximab⁵¹) while others such as Ustekinumab target *IL12A* and are licenced for several diseases (psoriasis⁵², psoriatic arthritis⁵³ and Crohn’s disease⁵⁴) and are being repurposed for other immune diseases such as systemic lupus erythematosus.⁵⁵ This vignette may help illustrate how the availability of GWAS results for mouth ulcers could facilitate repurposing of existing drug interventions or the development of novel, specific interventions for mouth ulcers.”

References:

1. Shi, H., Mancuso, N., Spendlove, S. & Pasaniuc, B. Local Genetic Correlation Gives Insights into the Shared Genetic Architecture of Complex Traits. *Am. J. Hum. Genet.* **101**, 737–751 (2017).
2. Rekaya, R., Smith, S., Hay, E. H. & Aggrey, S. E. Misclassification in binary responses and effect on genome-wide association studies. *Poult. Sci.* **92**, 2535–2540 (2013).

Review tables can be found in the excel file sent alongside this response and review figures are at the end of this document.

Review figure 1: Local genetic correlation between mouth ulcers and depressive symptoms. The top two panels show local genetic correlation and covariance for approximately 1700 loci genome-wide. The bottom two panel shows local heritability of mouth ulcers (panel 3, with peaks coinciding with lead signals on Manhattan plot) and depressive symptoms (panel 4).

Review figure 2: Local genetic correlation between mouth ulcers and neuroticism. The top two panels show local genetic correlation and covariance for approximately 1700 loci genome-wide. The bottom two panel shows local heritability of mouth ulcers (panel 3) and neuroticism (panel 4).

Reviewer #1 (Remarks to the Author):

The authors have satisfactorily responded to my comments and criticisms.

There is one typo I noted that should be corrected in the manuscript text. Results paragraph 5 states the p value for rs11928736 is 7.7 e-12 and should be 2.6 e=60 according to Table 2.

Reviewer #2 (Remarks to the Author):

No further comments. All previous comments were well addressed.

Response to reviewers:

Reviewer #1:

There is one typo I noted that should be corrected in the manuscript text. Results paragraph 5 states the p value for rs11928736 is 7.7 e-12 and should be 2.6 e=60 according to Table 2.

We have amended the p-value for rs11928736 to 2.6e-60 to match the correct value in Table 2.